# Analysis of a Sports-Educational Program in Prisons

**DOI:** 10.3390/ijerph17103467

**Published:** 2020-05-15

**Authors:** Gema Ortega Vila, Manuel Tomás Abad Robles, José Robles Rodríguez, Luis Javier Durán González, Jorge Franco Martín, Ana Concepción Jiménez Sánchez, Francisco Javier Giménez Fuentes-Guerra

**Affiliations:** 1Real Madrid Foundation, 28055 Madrid, Spain; gortega.fundacion@ext.realmadrid.es (G.O.V.); jfranco.fundacion@ext.realmadrid.es (J.F.M.); 2Faculty of Education, Psychology and Sport Sciences, University of Huelva, 21071 Huelva, Spain; jose.robles@dempc.uhu.es (J.R.R.); jfuentes@uhu.es (F.J.G.F.-G.); 3Faculty of Physical Activity and Sport Sciences, Polytechnic University of Madrid, 28040 Madrid, Spain; javier.duran@upm.es (L.J.D.G.); anaconcepcion.jimenez@upm.es (A.C.J.S.)

**Keywords:** prison, football, social reintegration

## Abstract

The aim of this study was to analyze the implementation of a sports-educational program in prisons where the Real Madrid Foundation’s Social-Sports Program is carried out. For that purpose, a survey of 468 inmates was conducted at 21 prisons in Spain where the Real Madrid Foundation implements its program (441 men (94.23%) and 27 women (5.77%)). Inmates stated that the program had had a favorable influence on their life in prison and considered that their participation in the program might have a significant influence on the likelihood of them continuing to play sports when they are released. In addition, inmates felt that they had learned a great deal about the contents relating to football and the values they had worked on, and that the program had had a very positive effect on their overall development as a person.

## 1. Introduction

The Spanish Prison Administration has clearly defined its purposes in Article 1 of the General Organic Law of the Penitentiary, which can be summarized as follows: Re-education and social reintegration of those sentenced to prison sentences and penal measures; retention and custody of detainees, prisoners, and convicts; and assistance and aid to inmates and released prisoners. The fundamental aim of this institution is the re-socialization of its inmates with the objective of reincorporating them into society by being genuinely free as citizens. To this end, there are many interventions and treatment programs that have been developed in prisons, including sports programs. These sports programs offered in prisons aim to “promote, through the practice of sport, attitudes, skills and behaviours that help to prevent high-risk social behaviour and favor insertion, trying to meet the existing demand and help to achieve physical, mental and social well-being” [1] (p. 83).

In this sense, the importance of physical exercise and sport for prisoners’ reintegration has been noted in several studies. Special mention should be given to the publications by Ríos [2]; Chamarro, Blasco, and Palenzuela [3]; and Castillo-Algarra [4] on sport and prisoner re-entry, and the works of Martos-García, Devís-Devís, and Sparkes [5,6] on the different social implications of playing sport and doing physical activity. The paper by Martínez, Martín, Usabiaga, and Martos [7] on the benefits and obstacles for female prisoners that do physical activity, as well as the research by Lleixá and Ríos [8] that focuses on the impact of a physical education program on inmates and an assessment of the lessons learnt by university students when implementing it, should be highlighted. Other excellent studies include that published by Parker, Meek, and Lewis [9], in which they looked at the use of sport in the rehabilitation of young people in England, and the work of Konstantinakos [10], which analysed the attitudes of Greek prisoners towards exercise and sport. Furthermore, Meek and Lewis [11] examined the promotion of health and well-being achieved by playing sports in prisons in England and Wales and the positive impact of sports programs on the reintegration of English prisoners [12]. Gallant, Sherry, and Nicholson [13] analysed the influence of sport on the rehabilitation of prisoners in Australia. Furthermore, Battaglia et al. [14] examined the impact of physical exercise on the psychological well-being of Italian inmates, while Pérez-Moreno et al. [15] researched the influence of physical exercise on the fitness of Spanish prisoners. Therefore, the practice of physical activity and sport occupies a prominent place in the life of prisons [3] and its importance in the rehabilitation of inmates has been recognized in various studies [3,9], although it is critical that its implementation be properly planned [9].

In short, it is clear that, when properly oriented, sport can aid in a person’s overall formation [16] because, as inmates taking part in sports programs themselves acknowledge, they not only learn technical-sporting aspects, but also personal and social values that facilitate their future integration into society [17,18]. Although the values that sport transmits to those that play it are highly heterogeneous [19,20,21,22], in the “Football at Penitentiary Centres” program, the Real Madrid Foundation (RMF) has focused on seven basic values: respect, autonomy, motivation, equality, self-esteem, health, and companionship. These are useful values in any environment, but especially in prisons. In addition, specialist literature emphasizes the need for such programs to be carried out in partnership with sports organisations and clubs from outside the prison [23], which is the case of the RMF. Moreover, it is important to explore inmates’ perceptions of sports programs, in order to assess their potential and impact [9,23].

In Spain, according to data from the General Secretariat of Penitentiary Institutions [1], the prison population consists of 50,461 inmates (46,669 males and 3792 females). Therefore, nine out of ten inmates are men (92.5%). Of the total, one in four inmates has a non-Spanish nationality (25.6%). In terms of age, these data reflect that one third of the prisoners are between 31 and 40 years old and another third are between 41 and 60. In total, 8.67% of them are between 21 and 25 years old and the youth population (less than 21 years old) is residual (0.5%). In addition, men are mainly in prison for the following offences: crimes against property and socio-economic order (36. 2%) and against public health (20. 1%). Crimes related to gender violence (9.5%) and homicide (7.5%) and against sexual freedom (6.7%) represent less of the offences. Together, the five crime categories represent 80.0% of the total crimes. On the other hand, female inmates are mainly in prison for crimes against property and socio-economic order (34.6%) and against public health (32.8%).

In this regard, few studies have evaluated the opinions of inmates regarding the implementation of sports programs in prisons. That is why the aim of this study was to analyze the implementation of a sports-educational program in prisons where the RMF’s Social-Sports Program is carried out.

## 2. Materials and Methods

### 2.1. Participants

The research centered on 468 prisoners (441 men (94.23%) and 27 women (5.77%)), of which 14 were less than 20 years old, 230 were aged between 20 and 30, 154 were between 31 and 40, 63 were between 41 and 50, and 6 were over the age of 50. In addition, 185 had a Spanish nationality, 233 had a foreign nationality, 25 had both a Spanish and foreign nationality, and 25 had a dual foreign nationality. Of the respondents, 79 inmates had been sentenced to less than 2 years’ incarceration, 141 to between 2 and 5 years, 105 to between 5 and 8 years, 64 to between 8 and 11 years, and 79 to more than 11 years. As far as education is concerned, 67 had no schooling, 203 had a primary education, 50 had received vocational training, 105 had a secondary education, and 44 had university degrees. The survey was conducted at the end of the program in the third quarter of the 2018/19 season. The respondents were participants in the Social-Sports football program run by the RMF at the following penitentiary centers in Spain: Algeciras, Aranjuez, Ávila, Castellón, Estremera, Ibiza, León, Meco I, Meco II, Murcia, Navalcarnero, Ocaña I, Ocaña II, Palencia, Segovia, Sevilla II, Soto, Teixeiro, Topas, Valdemoro, Villabona, and Zuera. In total, 195 of the inmates were taking part in the program for the first time, 148 for the second time, 62 for the third time, and 60 for the fourth time.

### 2.2. Instrument

The instrument used was the questionnaire designed and validated by Ortega et al. [24], which contemplates 70 items divided into dimensions: sociodemographic variables (6), attendance within the sports program (24), satisfaction with the experience (7), and results of the sports program (33). When preparing the items of the questionnaire, a four-point Likert scale was used, namely from 0 (no/not at all) to 3 (a lot). In addition, multiple choice questions (2, 3, and 4 alternatives) were included. The reliability of the questionnaire was evaluated by analysing its internal consistency using Cronbach’s Alpha method: 0.876 for the attendance within the sports program dimension, 0.565 for the satisfaction with the experience dimension, and 0.910 for the results of the sports program dimension, while the overall Cronbach’s Alpha value for the questionnaire was 0.930.

### 2.3. Procedure

The research was carried out in the third quarter of the 2018/19 season, after the necessary permits had been obtained from the various prison administrations. All ethical considerations involved in performing research with the participants were approved by the Training Committee of the Real Madrid Foundation, which also conformed to the recommendations of the Declaration of Helsinki. Additionally, all participants were informed about the research procedures and provided prior informed consent.

The coordinators of the football program at the prisons distributed the questionnaires among the inmates, after explaining to them the reasons for developing it and what it hoped to achieve, and also that participating in the survey was completely voluntary and that they could quit it whenever they liked. In all cases, inmates signed an informed consent form. In the program, football is played for one hour once a week and tries to use the sport as an entertaining, recreational, and educational activity. It also attempts to promote values that can be transferred to inmates’ life both inside the prison and in their future re-entry into society. In parallel, promoting healthy sport routines and suitable ways of resolving conflicts is also an important objective of the activity. The Social-Sports Program concludes by holding an in-house competition that aims to reinforce everything that has been achieved in training.

### 2.4. Statistical Analysis

A descriptive analysis of the central tendency and dispersion of items that comprise the questionnaire was carried out. The results of Kolmogorov–Smirnov normality testing showed the need for non-parametric tests (*p* < 0.05). In addition, the Mann-Whitney *U* test was used to determine whether significant differences based on gender existed. For intergroup comparisons (differences in age groups, academic training, and time spent playing sports in the program), the Kruskal–Wallis test by ranks was used. In cases where the intergroup comparisons were significant (*p* < 0.05), a pairwise comparison was made with the Mann-Whitney *U* test and Bonferroni adjustment to correct the significance level, and the size of the effect was calculated [25]. The SPSS 21.0 statistical analysis program for Windows was used for all of the analyses.

## 3. Results

### 3.1. Attendance within the Sports Program

Among the more prominent results of this dimension, it should be noted that most respondents joined the sports program because they liked playing sports (M = 2.75 ± 0.519) and because they considered it was good for their health (M = 2.62 ± 0.597), although other reasons also stood out. On the other hand, less importance was given to reasons involving interrelating with people outside the prison (M = 1.86 ± 1.410) and getting prison benefits (M = 1.50 ± 1.163) (see Table 1).

The Mann-Whitney U test revealed significant differences in the importance prisoners gave to joining the sports program as a way of relating to other inmates. This reason was of greater importance to women (M = 2.33 ± 0.920) than to men (M = 2.01 ± 0.898) (Z = −2.001, *p =* 0.045, r = −0.09), and the size of the effect was small. Regarding the reasons why they continued to attend the sports program training session, it should be noted that, as before, most respondents stated that they did so mainly for health reasons (M = 2.63 ± 0.584) and because they liked playing sports (M = 2.74 ± 0.531) (see Table 2).

From the Mann-Whitney U test, significant differences were found in the importance that prisoners attached to attending the sports program because that led to prison benefits. This reason was more important for men (M = 1.54 ± 1.186) than for women (M = 1.04 ± 1.192) (Z = −2.106, *p =* 0.035, r = −0.09), although the size of the effect was found to be small. On the other hand, significant differences (X^2^ = 7.994, *p =* 0.046) were detected in the importance of attending the sports program to get out of the pavilion between those who were taking part for the first time (M = 2.10 ± 0.961) and those who had been involved in the program for three seasons (M = 2.43 ± 0.921) (Z = −2.787, *p =* 0.005, r = −0.17), although the size of the effect was also seen to be small. Those who had been taking part in the sports program for longer attached a greater importance to getting out of the pavilion as a reason for attending the sessions.

### 3.2. Satisfaction with the Experience

In this dimension, it is well worth noting that most of the respondents were fairly or very satisfied with the RMF football program (M = 2.59 ± 0.598). It was also relevant that they expressed great satisfaction with the coach (M = 2.65 ± 0.573) and their playmates (M = 2.40 ± 0.628). However, they were less satisfied with how often the activity took place (M = 1.58 ± 0.875) and with how long the program lasted (M = 0.83 ± 0.489), so much so that the vast majority considered that the sports program should run for longer.

### 3.3. Results of the Sports Program

Respondents stated they had learned between ‘quite a lot’ and ‘a lot’ from the contents relating to football (M = 2.33 ± 0.683) and also in regard to the values that the Real Madrid Foundation sports program works on (see Table 3). Furthermore, in regard to their overall satisfaction with the work carried out by their coach, the inmates reported a very high or excellent degree of satisfaction. On the other hand, when respondents were asked about the influence that their participation in the sports football program had had on their prison life, it should be noted that the majority considered it had been favorable or highly favorable.

When each item in the dimension was compared by sex, Mann-Whitney revealed statistically significant results, although the size of the effect in all cases was low. Women (M = 2.70 ± 0.465) considered that they had learnt more than men had (M = 2.36 ± 0.600) about the values of companionship (Z = −2.930, *p =* 0.003, r = −0.13), self-esteem (Z = −3.120, *p =* 0.002, r = −0.14) (women M = 2.78 ± 0.424; men M = 2.39 ± 0.649), respect (Z = −2.390, *p =* 0.017, r = −0.11) (women M = 2.78 ± 0.424; men M = 2.56 ± 1.578), equality (Z = −2.295, *p =* 0.022, r = −0.10) (women M = 2.70 ± 0.542; men M = 2.43 ± 0.633), health (Z = −2.525, *p =* 0.012, r = −0.12) (women M = 2.85 ± 0.456; men M = 2.60 ± 0.563), motivation (Z = −2.625, *p =* 0.009, r = −0.12) (women M = 2.85 ± 0.362; men M = 2.56 ± 0.598), and autonomy (Z = −4.055, p < 0.001, r = −0.19) (women M = 2.89 ± 0.320; men M = 2.38 ± 0.688). In addition, inmates considered that their participation in the program was fairly or very influential on their physical condition (M = 2.37 ± 0.647), on their mental and emotional state (M = 2.37 ± 0.657), and on their relationships with other inmates (M = 2.29 ± 0.718). However, women (M = 3.78 ± 0.424) stated that their participation in the sports program had had a greater influence on their life in prison than for men (M = 3.36 ± 0.814) (Z = −3.09, *p =* 0.002, r = −0.14), on their mental and emotional state (women, M = 2.70 ± 0.465; men, M = 2.35 ± 0.662 with Z = −2.798, *p =* 0.005, r = −0.13), on their relationships with other inmates (women, M = 2.56 ± 0.641; men, M = 2.27 ± 0.720 with Z = −2.141, *p =* 0.032, r = −0.01), and on their relationships with the prison staff (women, M = 2.22 ± 1.086; men, M = 1.54 ± 1.093 with Z = −3.182, *p =* 0.001, r = −0.15). The size of the effect in all cases was low. In addition, respondents considered that their participation in the football program had helped them to have fun, to be happier, and to learn to make an effort (see Table 4).

When each item in the dimension was compared by sex, Mann-Whitney revealed statistically significant results. Women (M = 2.78 ± 0.424) estimated that their participation in the sports program was of greater help for them than for men (M = 2.29 ± 0.650) to learn to accept and abide by the rules (Z = −3.182, *p =* 0.001, r = −0.14), to respect others (women, M = 2.67 ± 0.555; men, M = 2.38 ± 0.603 with Z = −2.586, *p =* 0.010, r = −0.12), to learn to make an effort (women, M = 2.85 ± 0.362; men, M = 2.50 ± 0.588 with Z = −3.162, *p =* 0.002, r = −0.14), to appreciate their progress (women, M = 2.81 ± 0.396; men, M = 2.53 ± 1.587 with Z = −2.956, *p =* 0.003, r = −0.13), to make decisions (women, M = 2.78 ± 0.506; men, M = 2.34 ± 0.692 with Z = −3.484, *p* < 0.001, r = −0.16), to understand others better (women, M = 2.56 ± 0.751; men, M = 2.33 ± 0.702 with Z = −2.036, *p =* 0.042, r = −0.01), not to use drugs and tobacco (women, M = 2.52 ± 1.087; men, M = 2.08 ± 1.076 with Z = −2.781, *p =* 0.005, r = −0.13), and to have fun (women, M = 2.85 ± 0.456; men, M = 2.65 ± 0.528 with Z = −2.252, *p =* 0.024, r = −0.10), with the size of the effect being slight in all cases. On the other hand, respondents considered that their participation in the football program could bear greatly on their continuing to play sports when they were released from prison and on their acquisition or learning of positive values, on their overall development as a person, and on their future integration into society (see Table 5).

According to the Mann-Whitney U test, women (M = 2.52 ± 0.802) stated that their participation in the sports program had influenced them to a greater extent than men (M = 2.16 ± 0.839) in regard to their future re-entry into society (Z = −2.528, *p =* 0.011, r = −0.12), in their overall development as a person (women, M = 2.56 ± 0.411; men, M = 2.27 ± 0.776 with Z = −2.011, *p =* 0.044, r = −0.01), and in their acquisition or learning of positive values (women, M = 2.74 ± 0.526; men, M = 2.40 ± 0.670 with Z = −2.804, *p =* 0.005, r = −0.13). When intergroup comparisons were made with the Kruskal−Wallis contrast test, significant differences were detected (X^2^ = 1.414, *p =* 0.034). A pairwise comparison using Mann-Whitney with Bonferroni adjustment revealed significant differences between the age groups of less than 20−30 years (M = 2.20 ± 0.828) and 41−50 years (M = 1.90 ± 0.856, Z = −2.578, *p =* 0.010, r = −0.15), with a low effect size. Younger inmates considered, to a greater extent, that their participation in the football program would have a considerable influence on their future re-entry in society.

## 4. Discussion

The aim of this study was to analyze the implementation of a sports-educational program in prisons where the WFR’s Social-Sports Program is carried out. In this sense, coinciding with data obtained by Castillo-Algarra [4] and Psychou et al. [26], inmates stated that their participation in the program had had a favorable effect on their life in prison. They considered that it had influenced their relationships with other inmates, their physical and bodily state, and their mental and emotional state, which is consistent with other research [7,8,9,14,23,27,28,29].

With regard to the influence of the football program on personal affairs, the prisoners considered that their participation in the program could have a positive influence on their playing sports when they leave prison, on acquiring values in their development as individuals, and on helping them in their future integration into society [7]. In this sense, Chapter III of the Penitentiary Regulations (RD 190/1996) [30], which deals with training, culture, and sport in penitentiary institutions, acknowledges the role that socio-cultural and sports activities play as a means of promoting inmates’ overall personal development in the sense that they can provide benefits for convicts, such as permitting contact with outsiders, breaking with daily routines, and escaping from reality. Meek and Lewis [12] highlighted that sport has the potential to stimulate inmates to participate in reintegration programs, provides them with benefits within prison life and culture, and prepares them for life on the outside, and is thus an invaluable support for re-entry, although careful and deliberate planning of the practice is essential [9,23,29].

Physical exercise and sport can promote behavioral patterns related to healthier lifestyles for inmates and should occupy a prominent place in prisons [3]. Among the main reasons why inmates joined and attended the Sports Program were those connected with their desire to play sport and health. This data is consistent with the findings of other studies, such as Chamarro et al. [3], Konstantinakos [10], Ortega et al. [18,31], and Martín-González et al. [28]. In this sense, there are studies that suggest that the practice of physical exercise and sport produces health benefits for prisoners [13,32,33]. In addition, prisoners perceive that playing sports has a positive effect on their health [7,18,31,34,35]. All of the above highlights the importance of implementing physical activity and sports-based programs in prisons.

On the other hand, it is important to take into account the opinions of inmates regarding the implementation of physical activity and sport in prisons, in order to obtain better results [9,23]. Therefore, it is also worth mentioning that measuring user satisfaction is a suitable means of ascertaining the quality of an intervention program [36], and that an experience of satisfaction is a very important factor, since it is an excellent predictor of adherence to the program [37,38]. In this regard, it is significant that most respondents claimed to be very satisfied, which is highly relevant, because such a degree of satisfaction can encourage them to continue the activity. Lleixá and Ríos [8] concluded their study by stating that the prison population regards this type of experience as very positive and suggests that they should continue. Furthermore, inmates stated that they were highly satisfied with the work performed by the coach, which coincides with Garcia’s data [27]. Therefore, they demand that professionals run the physical exercise and sports programs in prisons [35].

In general, society considers that sport builds values in those who play it [23]. This is also the case among personnel working in penitentiary centres [17]. However, this transmission of values should be subordinate to the guidance given to playing sports, depending on which educational values may be transmitted [39]. For this reason, in the correctional context, it is only through a program with objectives and methodologies specifically designed to promote values that (re-)education can be achieved [4,6,40]. The results of the research carried out revealed that respondents claimed to have greatly learned about the contents specifically concerning football and also the underlying values covered [27,39]. Finally, inmates considered that the program had encouraged them to have fun and be happier and, to a lesser extent, not to use drugs and tobacco [4]. Besides, the inmates highlighted that their participation in the football program could have a significant influence on their continuing to play sport when getting out of prison, on the acquisition or learning of positive values, on their integral formation as a person, and on their future integration into society, which is in line with the findings of Martínez et al. [7].

The main limitation of our research refers to the scarcity of studies on inmate satisfaction with internal physical exercise and sport programs that reflect their opinions about how such programs are implemented. It should also be mentioned that, at the outset, there was some difficulty in distributing the questionnaire to people who did not fully understand Spanish, which was finally overcome by another inmate translating. Another limitation relates to the fact that no information was collected on the length of time that inmates had been in prison and its possible relationship to the results of the program. Therefore, more research on this aspect is needed. Besides, future research should focus on studying what type of physical exercise and which sports are most effective and most widely accepted in prisons. It would also be interesting to define, describe, and resolve the difficulties of implementing and operating sports programs and subsequently analysing them.

## 5. Conclusions

In conclusion, according to the results of this study, it is significant that inmates considered that their participation in the program has had a favorable influence on their life in prison and has helped them, in particular, to have fun and be happier. It should also be noted that inmates estimated that their participation in the football program positively influences the creation of the habit of playing sports, the acquisition of educational values, and their future integration in society. Furthermore, inmates expressed great satisfaction with the work performed by the coach, which is decisive for the continuity of the program.

These conclusions show that understanding what inmates think about doing physical exercise and sport is decisive when assessing the importance of this type of program in prisons. They also help to underline the importance of the proper planning and implementation of such programs and to recognize the need for further research on the subject.

## Figures and Tables

**Table 1 ijerph-17-03467-t001:** Reasons respondents gave for joining the sports program.

Items	Mean	Standard Deviation
Because it is good for my health	2.62	0.597
To care for/improve my physical appearance	2.40	0.703
To make friends with other inmates	2.03	0.901
To interact with people from outside the prison	1.86	1.410
To get prison benefits	1.50	1.163
To help time go by (a distraction/something to do)	2.37	0.811
To get out of the pavilion	2.12	1.005
Because I thought it would be fun	2.44	1.235
Because I like playing sports	2.75	0.519
To take part in a sports competition	2.51	0.733
To let off steam	2.32	0.845
How often did you go to the training session?	1.71	0.567

**Table 2 ijerph-17-03467-t002:** Importance respondents gave for continuing to attend the sports program.

Items	Mean	Standard Deviation
Because it is good for my health	2.63	0.584
Because it helps me care for/improve my physical appearance	2.45	0.719
Because I make friends with other inmates	2.11	0.872
Because I inter-relate with people from outside the prison	1.86	1.050
Because I get prison benefits	1.51	1.91
Because it helps time go by (a distraction and something to do)	2.35	0.812
Because that way I get out of the pavilion	2.18	0.935
Because I think it is fun	2.48	0.694
Because I like playing sport	2.74	0.531
Because it allows me to take part in a sports competition	2.51	0.801
Because I am able to let off steam	2.40	0.808

**Table 3 ijerph-17-03467-t003:** Respondent’s learning of the values worked on in the program.

Items	Mean	Standard Deviation
Overall, how much have you learnt about football?	2.33	0.683
Companionship	2.38	0.598
Self-esteem	2.41	0.644
Respect	2.57	1.535
Equality	2.45	0.631
Health	2.62	0.560
Motivation	2.57	0.590
Autonomy	2.41	0.683

**Table 4 ijerph-17-03467-t004:** Influence of the football program in other aspects.

Items	Mean	Standard Deviation
To learn to accept and abide by the rules	2.32	0.648
To respect others	2.39	0.603
To respect common materials and facilities	2.42	0.652
To learn to make an effort	2.52	0.583
To appreciate my progress	2.54	1.545
To belong to and collaborate with a group	2.52	0.588
To take decisions	2.37	0.690
To understand others better	2.34	0.706
Not to use drugs or tobacco	2.11	1.081
To have fun	2.66	0.526
To be happier	2.56	0.667

**Table 5 ijerph-17-03467-t005:** Influence of the football program on personal affairs.

Items	Mean	Standard Deviation
On my future integration into society	2.18	0.840
On my overall development as a person	2.28	0.762
On the acquisition or learning of positive values	2.42	0.667
On continuing to play sports when I leave prison	2.63	0.616

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
