# Peer review of "Analysis of a Sports-Educational Program in Prisons"

_ijerph, 2020, doi:10.3390/ijerph17103467_

Round 1

Reviewer 1 Report

I really enjoyed reading this manuscript. It is clear, well-written, and interesting. There are a couple of major questions I have about the research. 

1. The main thing that I continued questioning while reading the manuscript was the time period in which the prisoners took the questionnaire.

By that I mean I would like to see more information on the participants. Was the survey conducted at the beginning, middle, end of the program? How long had participants been in prison? I know you shared their sentences, but a key part of the questionnaire is considering the results of the program. I would think that individuals taking the questionnaire at different points in their sentence would have different perspectives on the results of the program. That's not the biggest deal to me if you don't have that information, but I do think it needs to be noted one way or another. 

2. Additionally, you present the purpose as being to "hear inmates' opinions about it." How was that accomplished? Did you interview participants?  I am a bit unclear about the phrasing. 

To that end, you focus the purpose statement heavily on satisfaction. However, satisfaction receives one paragraph in the results. I think the purpose statement needs to be reconsidered and that also leads to some issues with the discussion. 

3. Also related, you note that the participants "stressed that their participation in the sports programme could have significant influence on them continuing to play sport when released from prison, on the acquisition of positive values in their overall development as a person, and on their future integration into society." How was that stressed to you? Filling out a survey cannot provide some of the insights that are being focused on in the discussion. I think it is just the way it is presented by the authors and can be fixed. 

4. The larger concern is on the true aim of the paper. I read it as wanting to make a statement that the intervention works. But, we have no real understanding for how the program made a difference for inmates by way of comparison from some sort of pre post examination. In this case, a post then pre survey even seems appropriate. So, I understand in part why you focus on saying it is about satisfaction. But, that is not how the paper is presented. It needs to get worked out and made clearer what the point really is here. 

This is driven home by the fact that the most important conclusion that you present first is that the program had an influence on them and positively influenced habits. That is focused on results more so than satisfaction with the program. 

Author Response

COVER LETTER

Manuscript ID: ijerph-802748. Type of manuscript: Article. Title: Analysis of a Sports-Educational Programme in Prisons

Reviewer 1’s comments and suggestions for authors

Details of the revisions and responses

By that I mean I would like to see more information on the participants. Was the survey conducted at the beginning, middle, end of the program? How long had participants been in prison? I know you shared their sentences, but a key part of the questionnaire is considering the results of the program. I would think that individuals taking the questionnaire at different points in their sentence would have different perspectives on the results of the program. That's not the biggest deal to me if you don't have that information, but I do think it needs to be noted one way or another.

Information has been entered on the timing of the survey throughout the season. In addition, information is provided on the length of time prisoners have been in prison at the time of answering the survey, but no information is available on how long they have been in prison. However, this suggestion has been considered as a limitation of the study and is considered for future research.

Additionally, you present the purpose as being to "hear inmates' opinions about it." How was that accomplished? Did you interview participants?  I am a bit unclear about the phrasing.

To that end, you focus the purpose statement heavily on satisfaction. However, satisfaction receives one paragraph in the results. I think the purpose statement needs to be reconsidered and that also leads to some issues with the discussion.

This sentence has been removed from the text to avoid confusion.

We have reconsidered the purpose statement to avoid this question.

The aim of this study was to analyze the implementation of a sports-educational program in the prisons where the WFR's Social-Sports Program is carried out.

Also related, you note that the participants "stressed that their participation in the sports programme could have significant influence on them continuing to play sport when released from prison, on the acquisition of positive values in their overall development as a person, and on their future integration into society." How was that stressed to you? Filling out a survey cannot provide some of the insights that are being focused on in the discussion. I think it is just the way it is presented by the authors and can be fixed.

The recommendation is followed. We have modified the part of the text mentioned in order to be more precise. It sets out what inmates think and relates to the purpose attributed to sport by some prison laws and what some researchers consider to be the practice of sport in prisons.

With regard to the influence of the football programme on personal affairs, the prisoners considered that their participation in the programme could have a positive influence on their playing sports when they leave prison, on acquiring values in their development as individuals and on helping them in their future integration into societ

The larger concern is on the true aim of the paper. I read it as wanting to make a statement that the intervention works. But, we have no real understanding for how the program made a difference for inmates by way of comparison from some sort of pre post examination. In this case, a post then pre survey even seems appropriate. So, I understand in part why you focus on saying it is about satisfaction. But, that is not how the paper is presented. It needs to get worked out and made clearer what the point really is here.

We have reconsidered the aim of the research. This study aims to analyze the implementation of a social-sports program in relation to 3 dimensions: program attendance (e. g., reasons for attendance), satisfaction with the program experience and results (e. g., learning values).

THANK YOU for your comments and suggestions.

Reviewer 2 Report

The paper is well written and interesting, but it doesn't tell readers much about the inmate respondents, other than basic demographics.  In 250-500 words describe the prison system, security levels, housing (cell houses or ??), and conviction offenses of inmate respondents. Present data in Tables. How were data collected?  That's necessary for background.  Yes, inmates enjoy recreation, but does recreation reduce inmate offense frequency inside their respective prisons?  You must convince readers that inmate soccer players will be good, law-abiding citizens upon release.  I don't believe it. After all, if football players enjoyment had anything to do with crime, then lovers of football would not be in prison. 

Inmates playing cards or walking outside or watching television also enjoy those activities, but "so what?" I don't mean to overlook the excellent statistical analysis but that analysis has to be applied to more than inmate soccer play satisfaction.  Sport helps keep inmates in physical condition, which reduces health care costs inside prison.  That good to know; we know that about federal prisons in the U.S.  Do inmate sport participants commit fewer major offenses than non-participants while in prison?  Table 1 has two interesting finding, about attending class and prison benefits. Obviously prison won't reward inmates for playing soccer. After all the activity itself inside prison is a reward.  What does "class" mean?  Do inmate have jobs they are required to attend?  School are requirement? The response choices to survey questions are responses researchers thought were important.  I'd like to hear about soccer's down side, too.  Competition engenders anger.  Are there fights on the field?  Are these fights carried inside prison? And why were so few women interviewed?  Did mixed nationalities lead to violent aggression?   No amount of statistical analysis will convince me that prisoners will be good boys upon release, because they played soccer.  Perhaps soccer, housing, and food are benefits inmate footballers will enjoy if and when they return to prison.  I worked in a maximum-security federal penitentiary with about 1500 high-security inmates. Every inmate had a job; it was required. If inmates didn't work, they were locked up 24 hours a day.  Inmates earned money on their jobs.  I studied inmate work as its related to offenses levels.  I found that inmates who earned the most on their jobs did not commit violent offenses at the same frequency as inmates who earned less money.  Money was a gauge for education necessary to do particular jobs; inmates attended school to do jobs that paid the most money.  Very few ever committed a violent act. Low paid inmates--those who objected to the government's work requirement committed violent offenses quite often--they had invested little in their own well being.  Do have data on the number of hours inmates engage in football?  Can you link those activity hours to any disruptive behavior?  Your study could have been conducted in any institutional setting, schools? universities? Assembly lines in factories.  What makes this research interesting is its setting in prison.  So then, you must link football (hours per week; players prior to prison; aggression resulting from losing games), to daily life prison communities.   And, finally, your readers would enjoy reading inmate narratives about football as it relates to prison life. The research and this paper must give women a voice. 

Statistics is just another way of describing data. These data don't describe real-life behavior.  I would argue that if you sampled inmates who don't play football their responses would be similar to those who do play football. Why? Because there's a common, shared understanding of prison life among inmates, which includes learning how to think about football's role in prison life.   I'd bet if you sampled inmates in prisons you cited in the introduction, they too would share opinions about football in prison. 

When you rewrite this paper go easy on strings of statistics, which make reading rather unpleasant. At the very least, use put data in tables; and challenge your willingness to say that football is rehabilitative. What's the recidivism rate in Spain's prisons?  Yours is good work, worth publishing but you should go beyond what others have done. Ask hard questions of your data and your methods.

Author Response

COVER LETTER

Manuscript ID: ijerph-802748. Type of manuscript: Article. Title: Analysis of a Sports-Educational Programme in Prisons

Reviewer 2’s comments and suggestions for authors

Details of the revisions and responses

The paper is well written and interesting, but it doesn't tell readers much about the inmate respondents, other than basic demographics.  In 250-500 words describe the prison system, security levels, housing (cell houses or ??), and conviction offenses of inmate respondents. Present data in Tables. How were data collected?  That's necessary for background.  Yes, inmates enjoy recreation, but does recreation reduce inmate offense frequency inside their respective prisons?  You must convince readers that inmate soccer players will be good, law-abiding citizens upon release.  I don't believe it. After all, if football players enjoyment had anything to do with crime, then lovers of football would not be in prison.

Taking into account the Spanish prison legislation, the purposes of the Spanish prison system, the objectives and the role of the implementation of sports programs in Spanish prisons have been described.

The profile of the inmate population in Spanish prisons has also been described, as well as the reasons for the conviction of offenses of inmate

In addition, the section on materials and methods explains how the data were collected: the instrument subsection describes the questionnaire used to collect the data; the procedure subsection explains the process carried out to obtain the data; and the statistical analysis subsection describes the type of statistical analysis carried out and the statistics used.

This research does not seek to prove that playing football rehabilitates inmates. This study only intends to evaluate the implementation of a social-sports programme through the opinion of the participants, in this case, the inmates. For us, it is relevant what the interns think about the implemented program in relation to attendance, satisfaction and results.

What does "class" mean?  Does inmate have jobs they are required to attend?  School are requirement?

We've changed the term class to training session.

The response choices to survey questions are responses researchers thought were important.  I'd like to hear about soccer's down side, too.  Competition engenders anger.  Are there fights on the field?  Are these fights carried inside prison? And why were so few women interviewed?  Did mixed nationalities lead to violent aggression?

We find these questions very interesting and they will be taken into account for future research. The women surveyed were those who enrolled in the social-sports programme carried out.

No amount of statistical analysis will convince me that prisoners will be good boys upon release, because they played soccer.  Perhaps soccer, housing, and food are benefits inmate footballers will enjoy if and when they return to prison.  I worked in a maximum-security federal penitentiary with about 1500 high-security inmates. Every inmate had a job; it was required. If inmates didn't work, they were locked up 24 hours a day.  Inmates earned money on their jobs.  I studied inmate work as its related to offenses levels.  I found that inmates who earned the most on their jobs did not commit violent offenses at the same frequency as inmates who earned less money.  Money was a gauge for education necessary to do particular jobs; inmates attended school to do jobs that paid the most money.  Very few ever committed a violent act. Low paid inmates--those who objected to the government's work requirement committed violent offenses quite often--they had invested little in their own well being.  Do have data on the number of hours inmates engage in football?  Can you link those activity hours to any disruptive behavior?  Your study could have been conducted in any institutional setting, schools? universities? Assembly lines in factories. 

We agree with the reviewer that no one is going to become a good person by playing football. However, as the reviewer suggests, the education and welfare of prisoners is very important in reducing disruptive behaviours of prisoners. In this sense, the inmates stated that they signed up for the program for health and fun reasons, and that they had learned about sports content such as those related to values, and that all of this would serve them well in their prison life, and even outside of prison. However, it is true that this is what the inmates say, we do not really know the extent of their words in their real-life bahavior. Therefore, in this sense, more research is needed.

What makes this research interesting is its setting in prison.  So then, you must link football (hours per week; players prior to prison; aggression resulting from losing games), to daily life prison communities.   And, finally, your readers would enjoy reading inmate narratives about football as it relates to prison life. The research and this paper must give women a voice.

In our opinion, what is interesting about our study is to propose that sports practice (football in this case) in prisons, if deliberately planned and organized, can have educational benefits for inmates.

When you rewrite this paper go easy on strings of statistics, which make reading rather unpleasant. At the very least, use put data in tables; and challenge your willingness to say that football is rehabilitative. What's the recidivism rate in Spain's prisons?  Yours is good work, worth publishing but you should go beyond what others have done. Ask hard questions of your data and your methods.

Dear reviewer, we find these suggestions very interesting and stimulating, and they will be taken into account for future research.

Currently, we are conducting discussion groups with coaches and interns in order to deepen the data obtained in this work.

THANK YOU for your comments and suggestions.

Round 2

Reviewer 1 Report

I think the authors did a great job attending to the concerns. With the added contextualization upfront and the rephrasing of the purpose statement, I think the article is much clearer in design and makes a really powerful statement on rehabilitation in the prison system.